 SHORT REPORT

# Utility of estimated pulse wave velocity for assessing vascular stiffness: comparison of methods

**Stefan Möstl[1†], Fabian Hoffmann[1,2†], Jan-Niklas Hönemann[1,2], Jose Ramon Alvero-Cruz[3], Jörn Rittweger[4], Jens Tank[1], Jens Jordan[5,6]***

[1]Department of Cardiovascular Aerospace Medicine, Institute of Aerospace Medicine, German Aerospace Center, Cologne, Germany; [2]Department of Cardiology, University Hospital Cologne, Cologne, Germany; [3]Department of Human Physiology and Physical Sports Education, Faculty of Medicine, University of Málaga, Málaga, Spain; [4]Department of Muscle and Bone Metabolism, Institute of Aerospace Medicine, German Aerospace Center, Cologne, Germany; [5]Institute of Aerospace Medicine, German Aerospace Center, Cologne, Germany; [6]Chair of Aerospace Medicine, University of Cologne, Cologne, Germany

**\*For correspondence:**
Jens.Jordan@dlr.de

[†]These authors contributed equally to this work

**Competing interest:** The authors declare that no competing interests exist.

## Abstract

**Background:** Pulse wave velocity (PWV) independently predicts cardiovascular risk. Easy to use single-cuff oscillometric methods are utilized in clinical practice to estimate PWV. We applied the approach in master athletes to assess possible beneficial effects of lifelong exercise on vascular health. Furthermore, we compared single-cuff measurements with a two-cuff method in another cohort.

**Methods:** We obtained single-cuff upper arm oscillometric measurements thrice in 129 master athletes aged 35–86 years and estimated PWV using the ArcSolver algorithm. We applied the same method in 24 healthy persons aged 24–55 years participating in a head down tilt bedrest study. In the latter group, we also obtained direct PWV measurements using a thigh cuff.

**Results:** Estimated pulse velocity very highly correlated with age ($R^2 = 0.90$) in master athletes. Estimated PWV values were located on the same regression line like values obtained in participants of the head down tilt bed rest study. The modest correlation between estimated and measured PWV ($R^2$ 0.40; $p<0.05$) was attenuated after adjusting for age; the mean difference between PWV measurements was 1 m/s.

**Conclusions:** Estimated PWV mainly reflects the entered age rather than true vascular properties and, therefore, failed detecting beneficial effects of lifelong exercise.

**Funding:** The AGBRESA-Study was funded by the German Aerospace Center (DLR), the European Space Agency (ESA, contract number 4000113871/15/NL/PG), and the National Aeronautics and Space Administration (NASA, contract number 80JSC018P0078). FH received funding by the DLR and the German Federal Ministry of Economy and Technology, BMWi (50WB1816). SM, JT and JJ were supported by the Austrian Federal Ministry for Climate Action, Environment, Energy, Mobility, Innovation, and Technology, BMK (SPACE4ALL Project, FFG No. 866761).

## Editor's evaluation

This paper evaluates an algorithm for estimation of aortic pulse wave velocity (a robust measure of aortic stiffness), based on a simple, single-point (brachial artery) measurement. The authors test the algorithm using a number of approaches, including an interesting study of Masters athletes and

sedentary controls, where they identify a strong dependency of the algorithm on age, which has been documented previously. The paper, adds to the continuing debate over ideal methods for non-invasive estimation of aortic stiffness and will be of interest to other researchers working in the field of arterial hemodynamics, those interested in examining the health benefits of regular exercise and, possibly, health professionals looking for further tools to improve risk stratification.

## Introduction

Aortic pulse wave velocity (PWV) which relates to vascular stiffness, independently predicts cardiovascular risk including stroke (*Hametner et al., 2013*; *Vishram-Nielsen et al., 2020*). PWV values above 10 m/s have been suggested as threshold indicating increased risk (*Van Bortel et al., 2012*) and are included in current hypertension guidelines for assessing hypertension-mediated organ damage and guiding secondary prophylaxis (*Williams et al., 2018*). Local PWV measurements are also feasible using two points close or more distant to each other as well as measurements of pressure-velocity loops at one single spot. However, the guidelines do not specify certain methods to measure PWV. Measurements at different sites including the thigh were used in large scale studies. However, this PWV assessment is difficult to implement in busy clinics or during daily routine. Oscillometric blood pressure monitors estimating PWV based on age and blood pressure could simplify vascular assessment (*Hametner et al., 2013*; *Hametner et al., 2021*). Indeed, estimated PWV correlates with age (*Schwartz et al., 2019*) and invasively measured PWV (*Hametner et al., 2013*). These monitors operate observer-independent, are easy to use, and are, therefore, applied in clinical practice and in clinical studies alike (*Paiva et al., 2020*). However, in a study investigating beneficial effects of competitive lifelong physical exercise in elite Masters Athletes, we obtained estimated PWV results suggesting that the approach adds little information to classical cardiovascular risk assessment. We decided comparing the methodology with aortic PWV measurements in an independent set of healthy persons participating in a head-down-tilt bedrest study.

## Materials and methods

During the 23rd Masters Athletics Championships 2018 in Málaga, Spain, we approached 163 master athletes. We excluded athletes with atrial fibrillation or significant cardiovascular disease assessed by echocardiography. In the remaining 129 athletes (88 men/41 women, 56 ± 11 [range 35–86] years, 24.0 ± 3.5 kg/m²), we acquired blood pressure, heart rate, and estimated PWV thrice on the same arm after 10 min supine rest (Arc-Solver-Algorithm, CardioCube, AIT, Vienna, Austria).

In 24 healthy participants (16 men/8 women, 33 ± 9 years, 24.3 ± 2.1 kg/m²) of the AGBRESA (artificial gravity bedrest) study, which was conducted in collaboration with NASA and ESA in Cologne, Germany (*Hoffmann et al., 2021*), we assessed estimated PWV before head-down-tilt bedrest as described above. As the estimated PWV has been validated against invasive catheter measurements derived from the ascending aorta to the aortic bifurcation (*Hametner et al., 2013*), we mimicked this approach non-invasively by measuring pulse wave arrival time from the electrocardiographic R-peak to the arrival of the pulse wave at an oscillometric thigh cuff. We corrected pulse wave arrival time for isovolumetric contraction time, the time from the R-peak in the ECG to aortic valve opening assessed by 2D-pulsed-wave-doppler echocardiography. Corrected pulse wave arrival time represents pulse wave travel time from the aortic valve opening to the arrival at the thigh. Moreover, in contrast to the carotid-femoral PWV, this approach includes the ascending aorta and the aortic arc, which are commonly affected by aging-associated vascular disease. Dividing jugulum-thigh cuff distance by corrected pulse wave arrival time resulted in measured PWV.

All subjects provided informed consent and consent to publish before enrollment. The bedrest study as well as the study in master athletes was approved by the Northrine-Medical-Association (Ärztekammer Nordrhein, 2018143 and 2018171) ethics committee and registered at the German Clinical Trial Register (DRKS00015677 and DRKS00015172).

According to Kolmogorov-Smirnov test, all data were normally distributed, and results are reported as mean values ± standard deviation. Based on other cardiovascular risk estimate models, we used a quadratic regression model.

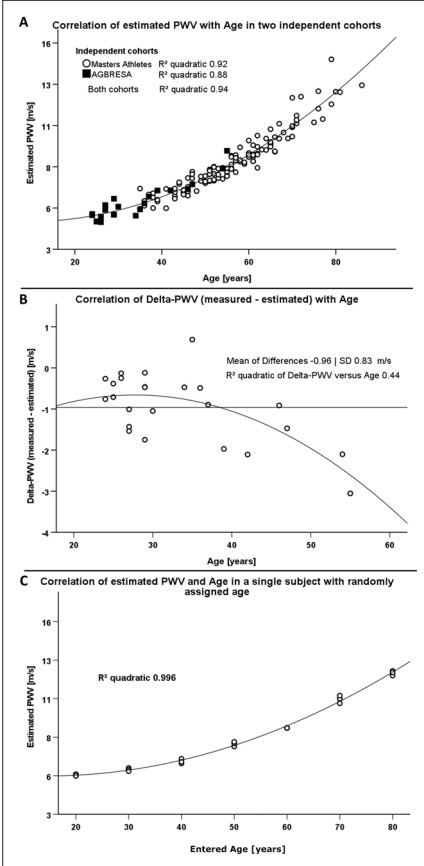

**Figure 1.** Comparison of two different methodologies to assess pulse wave velocity (PWV) in two different cohorts. (**A**) Correlation between estimated PWV and age in 129 Masters Athletes (empty circles) and in 24 AGBRESA study participants (black squares). (**B**) Regression analysis between the measured PWV and estimated PWV difference with age in 24 AGBRESA study participants. Estimated PWV deviated from measured PWV with increasing age (R² 0.44, p=0.07). (**C**) Quadratic regression analysis between estimated PWV and entered age in one man being in his 50s. We randomly entered age in decades from 20 to 80 and obtained measurements thrice at each entered age.

## Results

Resting heart rate was 61 ± 11 bpm, and blood pressure was 128 ± 15/78 ± 8 mmHg in master athletes. Estimated PWV ranged from 5.5 to 14.5 with a mean of 8.3 ± 1.8 m/s. In a quadratic regression model, age and mean arterial pressure predicted estimated PWV (beta value 0.93 and 0.14, p<0.001), whereas, sex and BMI had no influence. The model explained 95% (R² = 0.95) of estimated PWV's variance. Age alone explained 90% (R² = 0.90).

Resting heart rate was 62 ± 9 bpm, and blood pressure was 125 ± 11/70 ± 7 mmHg in the AGBRESA bedrest study. Estimated PWV was 5.8 ± 1.1 m/s, and measured PWV was 4.8 ± 0.6. Age and estimated PWV were highly correlated (R² 0.88, p<0.001, *Figure 1A*). The correlation between measured PWV and age was weaker (R² 0.55, p<0.001). In a quadratic regression model, age (beta value 0.99, p<0.001) but not BMI, sex, or mean arterial pressure predicted estimated PWV. The model explained 98% of estimated PWV's variance. R² between estimated and measured PWV was 0.40 (p<0.05). We observed an increasing bias between estimated and measured PWV with advancing age with an average 0.96 ± 0.83 m/s difference between methods (*Figure 1B*). When adjusting the correlation of measured and estimated PWV for age we could no longer observe a significant relationship (p=0.267).

To assess influences of entered rather than chronological age on estimated PWV, we repeated measurements in one male person, who is in his 50s and has normal BMI, and randomly changed the entered age in decades from 30 to 80 years. We obtained measurements in each entered age decade thrice. Again, estimated PWV related to entered age (R² 0.996, p<0.001, *Figure 1C*) akin to the relationship between estimated PWV and chronological age in athletes and in bedrest study participants.

## Discussion

The idea of obtaining vascular measurements, such as PWV, is gaining insight in individual vascular risk above and beyond traditional risk factors. The clinical goal is to target preventive measures to patients most likely to benefit. Similarly to other studies (*Schwartz et al., 2019*), we observed an almost perfect correlation between estimated PWV and age in two independent cohorts. Strikingly, estimated PWV in elite Master Athletes, which would be expected to benefit from lifelong exercise, and in healthy bedrest study participants of average physical fitness were located on the same regression line. Suffice it to say that many athletes were frustrated when we reported their findings.

Theoretically, our findings could result from methodological limitations of PWV estimates or a rather limited effect of lifelong exercise on aortic stiffness. A previous study showed significantly lower measured PWV in endurance trained compared with sedentary older men (*Vaitkevicius et al.,*

*1993*) making a methodological limitation more likely. Indeed, in our study, estimated PWV substantially overestimated PWV, particularly in older persons. Moreover, the modest correlation between estimated and measured PWV was attenuated after adjusting for age in our relatively small bedrest cohort. Finally, in a previous study, estimated PWV highly correlated with PWV calculated solely from age and blood pressure (*Hametner et al., 2021*). Thus, the algorithm providing PWV estimates appears to weigh age so strongly that subtle influences on vascular wall properties caused by hemodynamic changes and physical training cannot be discerned. Even more so, estimated PWV mainly reflects data entered prior to measuring blood pressure rather than true vascular properties. The clinical implication is that estimated PWV is no substitute for measured PWV, which likely limits the utility of the methodology in individualizing risk assessment. In fact, simply asking the patient in front of us for his or her age and measuring blood pressure provides almost as much information as estimating PWV. Instead, vascular aging assessment using established methodologies rather than estimates should be considered (*Jordan et al., 2015*).

## Acknowledgements

We would like to thank Siegfried Wassertheurer, Bernhard Hametner and Martin Bachler from the Austrian Institute of Technology for providing the CardioCube.

## Additional information

### Funding

| Funder | Grant reference number | Author |
| --- | --- | --- |
| German Federal Ministry of Economy and Technology | 50WB1816 | Fabian Hoffmann |
| Austrian Federal Ministry for Climate Action, Environment, Energy, Mobility, Innovation and Technology | FFG No. 866761 | Stefan Möstl Jens Tank Jens Jordan |

The funders had no role in study design, data collection and interpretation, or the decision to submit the work for publication.

### Author contributions

Stefan Möstl, Data curation, Formal analysis, Investigation, Methodology, Writing - original draft; Fabian Hoffmann, Data curation, Formal analysis, Visualization, Writing - original draft; Jan-Niklas Hönemann, Visualization, Writing - original draft; Jose Ramon Alvero-Cruz, Jörn Rittweger, Conceptualization, Project administration, Writing - review and editing; Jens Tank, Conceptualization, Project administration, Resources, Supervision, Writing - review and editing; Jens Jordan, Resources, Supervision, Writing - original draft

### Author ORCIDs

Stefan Möstl http://orcid.org/0000-0003-4983-4754
Jens Jordan http://orcid.org/0000-0003-4518-0706

### Ethics

All subjects provided informed consent and consent to publish before enrollment. The bedrest study as well as the study in master athletes were approved by the Northrine-Medical-Association (Ärztekammer Nordrhein, 2018143 and 2018171) ethics committee and registered at the German Clinical Trial Register (DRKS00015677 and DRKS00015172).

### Decision letter and Author response

Decision letter https://doi.org/10.7554/eLife.73428.sa1
Author response https://doi.org/10.7554/eLife.73428.sa2

## Additional files

### Supplementary files
• Transparent reporting form

### Data availability
As we obtained personal health data from human subjects, we cannot make their raw data publicly available. However, an interested researcher is able to access the original data by sending a project proposal to the corresponding author. This project proposal will be reviewed by the medical board of the DLR Institute of Aerospace Medicine. If the project is scientifically valuable, the committee decides to what extent the original data can be made available. Commercial research is excluded from this option. The syntax used for statistical analysis and the numerical data used to generate the figures are stored in Dryad: https://doi.org/10.5061/dryad.8931zcrsb.

The following dataset was generated:

| Author(s) | Year | Dataset title | Dataset URL | Database and Identifier |
|---|---|---|---|---|
| Möstl S, Hoffmann F, Hönemann J, Alvero-Cruz J, Rittweger J, Tank J, Jordan J | 2021 | Estimated pulse wave velocity is of limited utility to assess vascular stiffness | https://dx.doi.org/10.5061/dryad.8931zcrsb | Dryad Digital Repository, 10.5061/dryad.8931zcrsb |

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
