## [Editor Report]

This paper evaluates an algorithm for estimation of aortic pulse wave velocity (a robust measure of aortic stiffness), based on a simple, single-point (brachial artery) measurement. The authors test the algorithm using a number of approaches, including an interesting study of Masters athletes and sedentary controls, where they identify a strong dependency of the algorithm on age, which has been documented previously. The paper, adds to the continuing debate over ideal methods for non-invasive estimation of aortic stiffness and will be of interest to other researchers working in the field of arterial hemodynamics, those interested in examining the health benefits of regular exercise and, possibly, health professionals looking for further tools to improve risk stratification.

---

## [Decision Letter]

**Decision letter after peer review:**

Thank you for submitting your article "Utility of estimated pulse wave velocity for assessing vascular stiffness: comparison of methods" for consideration by *eLife*. Your article has been reviewed by 2 peer reviewers, and the evaluation has been overseen by a Reviewing Editor and a Senior Editor. The following individual involved in the review of your submission has agreed to reveal their identity: Jonathan P Mynard (Reviewer #2).

The reviewers have discussed their reviews with one another, and the Reviewing Editor has drafted this letter to help you prepare a revised submission.

Essential revisions:

1) Please address the following point by revising the text of your manuscript:

line 66: not all PWV measurements require a measurement on the thigh. Also, local measurements are feasible (2 points close to each other), as well as single-point measurements using two different biosignals (e.g. PU-loop method).

2) Please address the following point, both in your responses to both Reviewers and in revised text of your manuscript:

Why have you used PAT (corrected by isovolumic contraction time) from the heart to the thigh cuff to estimate PWV? When this reviewer understands the measurement setup correctly, another cuff was placed at the upper arm, which would allow PWV estimation using PTT.

*Reviewer #1 (Recommendations for the authors):*

line 66: not all PWV measurements require a measurement on the thigh. Also local measurements are feasible (2 points close to each other), as well as single-point measurements using two different biosignals (e.g. PU-loop method).

Why have you used PAT (corrected by isovolumic contraction time) from heart to the thigh cuff to estimate PWV? When this reviewer understands the measurement setup correctly, another cuff was placed at the upper arm, which would allow PWV estimation using PTT.

*Reviewer #2 (Recommendations for the authors):*

The authors used an unconventional (but acceptable) approach to measure aortic PWV, by using the ECG R-wave as the first timing marker, corrected for isovolumic contraction time. I suggest this approach could be made clearer, e.g. by stating that the time of aortic valve opening was used as the first timing marker, and this, in turn, was obtained from the ECG R-wave and echo-derived isovolumic contraction time.

---

## [Author Response]

Essential revisions:1) Please address the following point by revising the text of your manuscript:line 66: not all PWV measurements require a measurement on the thigh. Also, local measurements are feasible (2 points close to each other), as well as single-point measurements using two different biosignals (e.g. PU-loop method).

We addressed this point in the manuscript (line 64-66). Please also see our comments to the reviewers for more details.

2) Please address the following point, both in your responses to both Reviewers and in revised text of your manuscript:Why have you used PAT (corrected by isovolumic contraction time) from the heart to the thigh cuff to estimate PWV? When this reviewer understands the measurement setup correctly, another cuff was placed at the upper arm, which would allow PWV estimation using PTT.

We addressed this point in the manuscript (line 106-111). Please also see our comments to the reviewers for more details.

Reviewer #1 (Recommendations for the authors):line 66: not all PWV measurements require a measurement on the thigh. Also local measurements are feasible (2 points close to each other), as well as single-point measurements using two different biosignals (e.g. PU-loop method).

We responded to the comment and added a sentence regarding different approaches to measure PWV including single point measurements. However, large clinical trials showing the prognostic relevance of aortic or carotid femoral pulse wave velocity employed two point measurements at the carotid artery and femoral artery. Classical tonometric measurements were replaced later on by thigh cuff measurements mainly for practical reasons. We also appreciate the public review of reviewer 1, which comes to the conclusion that the role of different methodologies in estimating PWV remains to be shown. The major aim of our paper was to demonstrate that PWV estimates based on the ArcSolver algorithm exhibit major limitations in detecting changes induced by aging, bedrest, or lifelong physical activity. While some of these limitations have been previously recognized, blood pressure measurement devices incorporating such algorithms appear to promise more than they can keep. The recent literature suggests that clinicians and scientists who are no experts in PWV methodologies may not be aware of this limitation.

Why have you used PAT (corrected by isovolumic contraction time) from heart to the thigh cuff to estimate PWV? When this reviewer understands the measurement setup correctly, another cuff was placed at the upper arm, which would allow PWV estimation using PTT.

The reviewers are correct and we now explain more clearly (see also the comment to reviewer 2) our somewhat unconventional approach to use the R-peak of the ECG as first time marker corrected by the time interval until aortic valve opening (isovolumetric contraction time). Measuring the difference between the two arrival times of the pulse wave at the upper arm and at the thigh excludes the ascending and aortic arch part of the thoracic aorta, which may be relevant. Hence, we aimed with our approach to get a more complete picture comparable to the invasive catheter method or to the newly developed 4D-flow MRI measurements.

Reviewer #2 (Recommendations for the authors):The authors used an unconventional (but acceptable) approach to measure aortic PWV, by using the ECG R-wave as the first timing marker, corrected for isovolumic contraction time. I suggest this approach could be made clearer, e.g. by stating that the time of aortic valve opening was used as the first timing marker, and this, in turn, was obtained from the ECG R-wave and echo-derived isovolumic contraction time.

Please see the response to reviewer 1. We revised the description of our corrected PAT approach and stated the advantage of using the R-peak of the ECG corrected by the time to aortic valve opening (isovolumetric contraction time) to measure aortic pulse wave travel time including the ascending aorta and the aortic arch.